# Effect of Biodentine on Odonto/Osteogenic Differentiation of Human Dental Pulp Stem Cells

**DOI:** 10.3390/bioengineering10010012

**Published:** 2022-12-21

**Authors:** Xuerong Wang, Yixin Cai, Min Zhang, Junchen Xu, Chengfei Zhang, Jin Li

**Affiliations:** 1Jiangsu Province Engineering Research Center of Stomatological Translational Medicine, Jiangsu Province Key Laboratory of Oral Diseases, Department of Geriatric Dentistry, The Affiliated Stomatological Hospital of Nanjing Medical University, 136 Hanzhong Road, Nanjing 210029, China; 2Department of Endodontics, Yancheng Stomatological Hospital, Yancheng 224000, China; 3Restorative Dental Sciences, Faculty of Dentistry, The University of Hong Kong, Hong Kong 999077, China

**Keywords:** Biodentine, hDPSCs, vital pulp therapy, differentiation, senescence

## Abstract

This study aims to compare the biological characteristics of human dental pulp stem cells (hDPSCs) isolated from different-aged populations and examine the effects of Biodentine on proliferation and odonto/osteogenic differentiation of hDPSCs isolated from the elderly in vitro. hDPSCs were isolated from three different-aged populations: group A (≤18 years old), group B (19–59 years old), and group C (≥60 years old). The adhesion, proliferation, odonto/osteogenesis, and senescence were compared. The optimal concentration of aqueous Biodentine extract was determined by CCK-8 assay, alkaline phosphatase (ALP), and alizarin red staining (ARS). The effect of Biodentine on odonto/osteogenic gene and protein expression of hDPSCs in each group was evaluated by quantitative real-time PCR (QRT-PCR) and Western blot. hDPSCs were successfully isolated from three different-aged populations. Flow cytometry revealed that all isolated hDPSCs were positive for CD73 (>90%), CD90 (>90%), CD146 (<30%), and negative for CD45 (<1%). There existed an age-related decline in proliferation, odonto/osteogenic gene expression, and S-phase fraction (*p* < 0.05), an increase in senescence genes and p21 and p16 expression, and time needed for cell adhesion. Biodentine promoted hDPSC proliferation and mineralization in each group, particularly at a concentration of 0.2 mg/mL. Biodentine markedly enhanced odonto/osteogenesis-related gene and protein expression in each group (*p* < 0.05). hDPSCs can be obtained from populations of all ages. Though there is an age-related decline in their biological properties, hDPSCs from the elderly still maintain certain proliferation and multidirectional differentiation abilities. Biodentine can significantly promote the proliferation and odonto/osteogenic differentiation of hDPSCs isolated from the elderly over 60 years old, which could be considered a pulp capping material for vital pulp therapy in the elderly. Nevertheless, the efficacy of Biodentine in clinical application has to be further studied.

## 1. Introduction

With the promotion of minimally invasive dentistry, vital pulp therapy (VPT) has attracted considerable attention in clinical practice [1]. VPT is a minimal intervention strategy to protect the affected pulp from infection and partially preserve the living pulp tissues by excising the inflamed portion, which can maintain the vitality and physiological functions of the pulp [2,3]. Compared to root canal therapy, VPT minimizes the removal of tooth hard tissue, increasing tooth strength and fracture resistance to prolong the affected tooth’s service life [4].

The success of VPT depends on a variety of factors. The proliferation and odonto/osteogenic differentiation capacity of stem cells are critical [5]. Human dental pulp stem cells (hDPSCs) can be isolated from primary, immature permanent, and mature permanent teeth of young adults [6,7,8,9,10]. hDPSCs have received extensive attention in the field of tissue engineering and regenerative medicine due to accessibility and multilineage differentiation capacity [11,12]. hDPSCs can differentiate into several tissues including cartilage, bone, adipose, vascular and nervous tissues [13] and they have been extensively studied for the treatment of diseases such as bone defects, stress urinary incontinence, retinal degenerative disorders, and other diseases [14,15,16].

However, research on hDPSCs isolated from elderly permanent teeth is insufficient [9]. It has been reported that aging reduces the number of cells in the pulp and causes cell morphological changes and accumulation of collagen fibers, resulting in the degeneration of the dental pulp, which affects its vitality [17,18].

Pulp capping agents play a vital role in the efficacy of VPT [19]. Mineral trioxide aggregate (MTA) is widely used as the material of choice for pulp capping. However, concerns associated with MTA, such as tooth discoloration, lengthy setting time, and poor handling, still exist [20]. Biodentine is a bioactive dentin replacement material manufactured to overcome the disadvantages of MTA, exhibiting less discoloration, reduced setting time, enhanced handling and mechanical properties, and adequate radiopacity [21,22]. Furthermore, Biodentine has emerged as a pulp-capping material due to its excellent biocompatibility and physicochemical properties [23]. It can protect pulp nerves from external stimuli and promote the proliferation and odonto/osteogenic differentiation of hDPSCs [24,25]. Therefore, it is rational to speculate that this material may enhance the proliferation and differentiation of hDPSCs in the elderly, thereby improving the success rate of VPT in this group.

The present in vitro study aimed to investigate the biological properties of hDPSCs from different ages and examine the effects of Biodentine on the proliferation and odonto/osteogenic differentiation of hDPSCs in elderly patients over 60 years old.

## 2. Materials and Methods

### 2.1. Preparation of Biodentine

Biodentine (Septodont, Saint-Maur-des-Fossés, France) was mixed according to manufacturer instructions at room temperature and solidified for 2 h. Samples were ground into powder and irradiated with a UV lamp (365 nm) overnight. Afterward, the powder was added to alpha-modified Eagle medium (α-MEM, Gibco, Grand Island, NY, USA) with a ratio of 600 mg powder per 15 mL α-MEM and incubated at 37 °C for 72 h. After filtration through a 0.22 mm filter (Millipore, MA, USA), the concentration of the extracts was 40 mg/mL. The stock solution was diluted stepwise into four different concentrations, 20, 2, 0.2, and 0.02 mg/mL, and stored at 4 °C.

### 2.2. Biological Characteristics of hDPSCs 

#### 2.2.1. Isolation and Characterization of hDPSCs

The Institutional Ethics Committee of Nanjing Medical University approved this study (IRB No. PJ2020-094-001). Intact, extracted molars or premolars were collected from patients of different ages at Oral and Maxillofacial Surgery of the Jiangsu Provincial Stomatological Hospital after obtaining informed consent. Patients were divided into 3 groups according to Harms et al. [26] and age segmentation criteria recommended by China: group A (n = 6): ≤18 years; group B (n = 6): 19–59 years; and group C (n = 5): ≥60 years. Briefly, the teeth were cleaned and cut transversely. Afterward, the pulp tissue was removed gently from the dental pulp cavity in an aseptic manner and cut into small pieces. The tissue pieces were digested using 3 mg/mL type I collagenase (Gibco) with 4 mg/mL dispase (Gibco) for 1 h at 37 °C and cultured in α-MEM with 10% fetal bovine serum (FBS, ScienCell, Carlsbad, CA, USA) and 2% penicillin–streptomycin (NCM, Suzhou, China) in a humidified atmosphere containing 5% CO_2_ at 37 °C after centrifugation. Cell culture medium was refreshed every 3 days. Cell adhesion was characterized by phalloidin staining. Briefly, fixed cells were stained with fluorophore-conjugated phalloidin (APExBIO, Houston, USA) for 30 min, followed by DAPI (Beyotime, Shanghai, China) staining for 2 min. When 90% confluence was achieved, cells were digested using 0.25% trypsin (Gibco) and subcultured. hDPSCs from passages 2–4 were used in the following experiments.

hDPSCs at passage 2 were used to assess their stemness by flow cytometric analysis. The collected cells were rinsed with phosphate-buffered solution (PBS, Gibco), incubated with antibodies at 4 °C for 30 min, and analyzed by flow cytometry (BD, Franklin Lakes, NJ, USA). The following antibodies were used: CD34 FITC, CD90 APC, CD73 PE-Cy7, and CD146 PE (BD). 

#### 2.2.2. Proliferation Assay 

Cell proliferation was assessed by a Cell Counting Kit-8 (CCK-8, Beyotime, Shanghai, China). hDPSCs were seeded in a 96-well plate at a density of 1 × 10^3^ cells/well and incubated at 37 °C in 5% CO_2_ for 1 week. At days 1, 3, 5, and 7, 10 μL of CCK-8 solution was added to each well and incubated for 2 h. The optical density (OD) was measured at 450 nm using a plate reader (Spectramax, Sunnyvale, CA, USA).

#### 2.2.3. Quantitative Reverse Transcription Polymerase Chain Reaction (QRT-PCR)

Cells of different age groups were further divided into the osteoinduction group and the control group. Osteogenic induction medium containing 10 mmol/L β-glycerophosphate (Sigma, St Louis, MO, USA), 50 mg/L vitamin C (Solarbio, Beijing, China), and 10 nmol/L dexamethasone (APExBIO) was used in the osteoinduction group. In contrast, a complete culture medium was used as a control. Seven days after culturing, total RNA was extracted by a MiniBEST Universal RNA Extraction Kit (Takara, Tokyo, Japan). cDNA was synthesized by reverse transcription using PrimeScript™ RT Master Mix (Takara). Equal amounts of cDNA were used for real-time amplification of the target genes using ChamQ Universal SYBR qPCR Master Mix (Vazyme, Piscataway, NJ, USA). The reactions were performed on a QuantStudio^TM^ 7 Flex Real-Time PCR System (Applied Biosystems, Thermo Fisher Scientific, Waltham, MA, USA) at 95 °C for 30 s for one cycle, and then 95 °C for 10 s, 60 °C for 30 s for 40 cycles, with a final extension at 95 °C for 15 s, 60 °C for 1 min, and 95 °C for 15 s. The 2^—ΔΔCt^ method was applied to calculate the relative expression of odontogenic/osteogenic genes. Primer sequences are shown in Table 1.

#### 2.2.4. Cell Cycle Assay

The cell cycling phases were determined by flow cytometry. Briefly, cells of the three groups were collected and adjusted to 1 × 10^6^ cells. Then, the cells were washed 3 times with PBS. For cell cycle analysis, the cultured cells were fixed with 70% ethanol overnight at 4 °C. Next, the fixed cells were washed twice with PBS and stained with PI/RNase staining buffer (BD) at room temperature in the dark for 15 min.

#### 2.2.5. Western Blot Analysis

Western blot was performed to detect changes in senescence genes in hDPSCs from different ages. After being cultured in a complete culture medium for 2 weeks, hDPSCs in all groups were lysed with RIPA buffer (Beyotime) for protein extraction. Protein separation was carried out by polyacrylamide gel electrophoresis. The electrophoretic transfer method was used to transfer the protein to PVDF membranes (Millipore). The membranes were then blocked with skim milk at room temperature for 2 h and incubated with primary antibodies overnight at 4 °C. The following primary antibodies were used: anti-p21 (dilution 1:1000; Cell Signaling, MA, USA), anti-p16 (dilution 1:1000; Cell Signaling), and anti-GAPDH (dilution 1:1000; Proteintech, Chicago, IN, USA). Next, the membranes were incubated with secondary antibodies (dilution 1:10,000; Abcam, MA, USA) that corresponded with primary antibodies for 1 h. Finally, with the help of ECL chemiluminescent reagents (NCM), the protein bands were shown under a chemiluminescence imaging system (Tanon, Shanghai, China). The relative protein expression levels of senescence genes were measured using GAPDH as a reference protein.

### 2.3. The Influence of Biodentine on hDPSCs

#### 2.3.1. CCK-8 Assay

CCK-8 assay was performed to assess the influence of Biodentine on the proliferation of hDPSCs. hDPSCs of each age group were cultivated at a density of 1 × 10^3^/well in 96-well plates with extracts of Biodentine (0.02, 0.2, 2, and 20 mg/mL). CCK-8 was added and incubated with the cells for 2 h at days 1, 3, 5, and 7. The OD value at 450 nm was assayed (Spectramax).

#### 2.3.2. Alizarin Red Staining (ARS)

To study the effects of various concentrations of Biodentine extracts on mineralization of hDPSCs from different ages, hDPSCs were seeded at 1 × 10^5^ cells/well in a 6-well plate and treated with varying concentrations of Biodentine extracts at 90% confluency. After 2 weeks of induction, the cells were fixed with 4% paraformaldehyde (Beyotime) for 30 min, followed by staining with alizarin red solution (Solarbio) for 30 min at room temperature. Stained cells were observed under an inverted microscope (Leica, Wetzlar, Germany). For mineralization quantification, destaining was conducted by adding a 10% cetylpyridinium chloride solution (Solarbio). The eluant was blended and transferred to a 96-well plate. Absorbance was measured at 405 nm (Spectramax).

#### 2.3.3. Alkaline Phosphatase (ALP) Staining and ALP Activity Assay

To further evaluate the effects of Biodentine on the mineralization of hDPSCs from different ages, ALP staining and ALP activity assay were performed. hDPSCs were cultured, as shown in Section 2.3.2. ALP staining was conducted with the BCIP/NBT Alkaline Phosphatase Color Development Kit (Beyotime) at day 14 after induction according to the manufacturer’s instructions. Stained cells were then observed under an inverted microscope (Leica). The ALP activity was measured using an Alkaline Phosphatase Assay Kit (Jiancheng, Nanjing, China). The OD value was measured at 520 nm (Spectramax).

#### 2.3.4. QRT-PCR

According to the above experimental results, the optimal concentration of Biodentine extract (0.2 mg/mL) with no toxicity and the best ability to induce mineralization of hDPSCs in the three groups was chosen for subsequent experiments. Cells of different age groups were further divided into two groups: the Biodentine group and the control group. The Biodentine group was cultured with Biodentine extract, while the control group was cultured with a complete culture medium for 1 week at 90% confluency. Then QRT-PCR was used to detect the mRNA expression levels of odontogenic/osteogenic genes, dentin sialophosphoprotein (DSPP), osteocalcin (OCN), runt-related transcription factor 2 (RUNX2), and type I collagen (COL-1). The 2^−ΔΔCt^ method was applied to calculate the relative expression of odontogenic/osteogenic genes.

#### 2.3.5. Western Blot Analysis

hDPSCs were cultured, as shown in Section 2.3.4, for 2 weeks. Western blot was conducted to determine the protein expression of DSPP, OCN, RUNX2, and COL-1. The following primary antibodies were used: anti-DSPP (dilution 1:1000; Affinity Biosciences, OH, USA), anti-OCN (dilution 1: 1000; Abcam), anti-RUNX2 (dilution 1:1000; Cell Signaling), anti-COL-1 (dilution 1:1000; Cell Signaling), and anti-GAPDH (dilution 1:1000). The protein bands were detected by a chemiluminescence imaging system. 

### 2.4. Statistical Analysis

All experiments were performed independently at least 3 times, and the data are expressed as the mean ± standard deviation (SD). All statistical calculations were performed using SPSS Statistics 23.0 software (IBM, Armonk, NY, USA). One-way ANOVA was used for comparing multiple groups, and the means of two groups were compared using Student’s *t*-test. A value of *p* < 0.05 was considered statistically significant.

## 3. Results

### 3.1. Characterization of hDPSCs from Different Ages

hDPSCs of different ages were successfully isolated and cultivated by enzymatic digestion. Adherent cells could be observed under the microscope after 3–8 days of incubation. The time required for cell adhesion was prolonged with aging among the three groups (*p* < 0.05) (Figure 1) (Table 2). 

hDPSCs in all groups were spindle-shaped, with large cell bodies and nuclei located in the center of cells. After being passaged at the same density, hDPSCs in different groups were attached well. However, compared with the other two groups, some cells in group C displayed elongation and irregularities, and nuclear fragmentation appeared more frequently (Figure 2). According to the flow cytometric analysis, hDPSCs in these three groups were positive for CD73 (>90%), CD90 (>90%), with low expression of CD146 (<30%), and negative for CD45 (<1%) (Figure 3). The result suggested that the cultured cells had the characteristics of mesenchymal stem cells (MSCs).

### 3.2. Proliferation of hDPSCs from Different Ages

The proliferative potential displayed a declining trend with age. On day 7, the proliferation of hDPSCs entered a plateau phase. In the meanwhile, the proliferation level of group C decreased significantly compared with the other groups (*p* < 0.05), but there was no significant difference between group A and group B (*p* > 0.05) (Figure 4).

### 3.3. Odonto/Osteogenic Differentiation of hDPSCs 

The mRNA expression of DSPP, OCN, RUNX2, and OPN was significantly up-regulated (*p* < 0.05), apart from COL-1, compared with the control group (*p* > 0.05) in groups A and B. Meanwhile, in group C, increased mRNA was detected for the DSPP and OCN genes (*p* < 0.05), and decreased mRNA was detected for the OPN and COL-1 genes in comparison with the control group (*p* < 0.05), without statistical significance in the expression of RUNX2 (*p* > 0.05) (Figure 5).

### 3.4. Senescence of hDPSCs from Different Ages

Flow cytometric analysis revealed a significant decrease in the proportion of cells in the S phase with age (*p* < 0.05) (Figure 6a,b). The result showed that the expression of p21 and p16 increased with age (*p* < 0.05) (Figure 6c,d).

### 3.5. Effects of Biodentine on the Proliferation of hDPSCs 

The result displayed that Biodentine promoted the proliferation of hDPSCs in all groups at the concentrations of 0.02, 0.2, and 2 mg/mL (*p* < 0.05). However, Biodentine at the concentration of 20 mg/mL significantly inhibited proliferation in contrast to the control group (*p* < 0.05) (Figure 7). Hence, Biodentine at the concentration of 20 mg/mL was not used for the subsequent experiments.

### 3.6. Effects of Biodentine on the Mineralization of hDPSCs

hDPSCs from all groups cultured with different concentrations of Biodentine for 2 weeks showed more accumulation nodules compared with the control group. hDPSCs treated with 0.2 mg/mL Biodentine displayed the highest calcium concentration (Figure 8a,b). Furthermore, the result indicated that different concentrations of Biodentine extracts could significantly induce positive ALP staining, with the greatest staining seen in hDPSCs cultured with 0.2 mg/mL Biodentine (Figure 8c). In treatment with different concentrations of Biodentine, the ALP activity of the three groups increased markedly (*p* < 0.05), and hDPSCs cultured with 0.2 mg/mL Biodentine obtained the highest ALP activity (Figure 8d). Based on the above results, 0.2 mg/mL was used for the subsequent experiments.

### 3.7. Biodentine on Odonto/Osteogenic Differentiation of hDPSCs

Results showed that mRNA levels of DSPP, OCN, RUNX2, and COL-1 genes were up-regulated significantly (*p* < 0.05) compared with the control group (Figure 9a). Moreover, the result of Western blot indicated that protein expression of these odonto/osteogenic genes was significantly increased compared with the control group (*p* < 0.05) (Figure 9b,c).

## 4. Discussion

There has been an increasing need for dental care and treatment as the proportion of the older population in societies upsurges. Due to disabilities, oral health conditions of the elderly are generally poor, often leading to pulp infection or necrosis, severe periodontitis, and tooth extraction [27,28]. Continuous exposure of teeth to physiological and pathophysiological stimuli causes the formation of tertiary dentin, resulting in pulp chamber and root canal calcification [29]. In such a scenario, the volume and regenerative capacity of pulp tissue decrease, posing limitations in VPT in the elderly [30]. Furthermore, as shown in this study, it is difficult or impossible to obtain pulp tissue in some cases. In this study, we obtained five pulp samples (≥60 years old, with a maximum age of 76) from the eligible teeth, although the successfully isolated cells were positive for CD73 and CD90 (Figure 3). Meanwhile, we found a high percentage of expression of CD146 in group A compared to both group B and group C. Matsui et al. found that CD146^+^ cells may promote mineralization and generate dentin/pulp-like structures, suggesting a role in self-renewal of stem cells and dental pulp regenerative therapy [31], which is comparable to our findings. In addition, hDPSCs isolated from the elderly had certain proliferation and multilineage differentiation capacities. These results suggest that VPT could be applied to the teeth of the elderly.

Previous studies have compared the biological properties of hDPSCs and stem cells from exfoliated deciduous teeth (SHEDs) in vitro and in vivo and found no difference between them [32], suggesting that both deciduous teeth and permanent teeth can be treated by VPT. In order to investigate the changes in the biological characteristics of hDPSCs in the elderly, patients of different ages were studied. It was shown that the structure of some hDPSCs changed with the increase in age (Figure 2). There existed an age-related decline in the proliferation and odonto/osteogenic differentiation capacity (Figure 4 and Figure 5). It was also found that S phase decreased progressively with age (Figure 6a,b), indicating that the transformation of cells from G0/G1 to the S phase was inhibited. At the same time, p21 and p16 protein levels increased (Figure 6c,d). These signs of cellular senescence may owe to the dysregulation of antioxidant mechanisms in hDPSCs [33,34]. Since cellular senescence has a great influence on the repairing capacity of injured pulp, the proliferation and odonto/osteogenesis of hDPSCs were studied. Cheng et al. analyzed the overexpression of WNT10a in hDPSCs, which promoted cell proliferation by increasing G2/M- and S-phase cells via the canonical WNT/β-catenin signaling transduction [35]. Wnt signaling involves embryonic development, tissue homeostasis maintenance, and cellular activities. The canonical WNT/β-catenin pathway activated by the high stiffness of scaffolds induced odontogenic differentiation in hDPSCs [36,37]. Stem cells from the elderly, similar to cells from the young, exhibit a reaction to these signals [38,39]. Nevertheless, the increased oxidative stress caused by aging activates specific transcription factors that could translocate β-catenin, attenuating the conduction of this pathway and ultimately inhibiting osteogenesis [40]. Therefore, it is supposed that WNT/β-catenin signaling pathway inhibition may be correlated with aging-associated changes in functions of hDPSCs. 

Biodentine is a bioactive dentin replacement material, which is composed of tricalcium silicate, dicalcium silicate, calcium carbonate, etc. [23,41,42]. Studies have shown that Biodentine significantly promoted the proliferation and odonto/osteogenic differentiation of SHEDs and hDPSCs in young adults [25,43]. In this study, three age groups (≤18, 19–59, and ≥60 years old) were studied. The effects of Biodentine on hDPSCs isolated from the elderly clarify the application of this material in the preservation of vital pulp in the elderly. CCK-8 assay showed that different concentrations of Biodentine extracts could significantly promote the proliferation of each group, indicating that appropriate concentrations of Biodentine extracts were nontoxic (Figure 7). The results of alizarin red S and ALP staining showed that Biodentine had an obvious positive influence on the mineralization of hDPSCs in the three groups, especially the concentration of 0.2 mg/mL (Figure 8). qRT-PCR and Western blot further proved that Biodentine at this concentration could significantly increase the expression of odonto/osteogenic genes, such as DSPP, OCN, RUNX2, and COL-1 (Figure 9). The above experimental results showed that although significant differences among groups were identified, Biodentine could promote the proliferation and odonto/osteogenic differentiation of hDPSCs, suggesting that Biodentine could be used for VPT in the elderly on the basis of selecting appropriate cases and ensuring a sterile treatment environment.

## 5. Conclusions

In conclusion, this study revealed the biological characteristics of hDPSCs in the elderly and found that Biodentine promoted the proliferation and odonto/osteogenic differentiation of hDPSCs in elderly patients over 60. Biodentine is a valuable material for VPT in the elderly. However, the underlying mechanisms of Biodentine in regulating the differentiation of hDPSCs require further study.

## Figures and Tables

**Figure 1 bioengineering-10-00012-f001:**
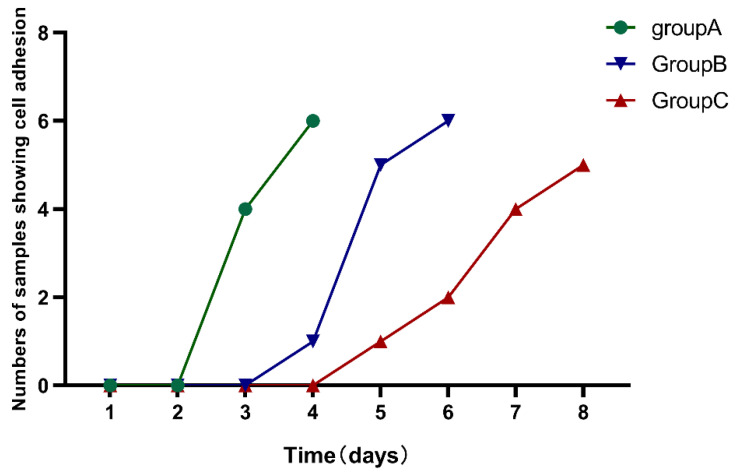
The time required for cell adhesion was observed among the three groups. The time required for cell adhesion was prolonged with aging in the three groups.

**Figure 2 bioengineering-10-00012-f002:**
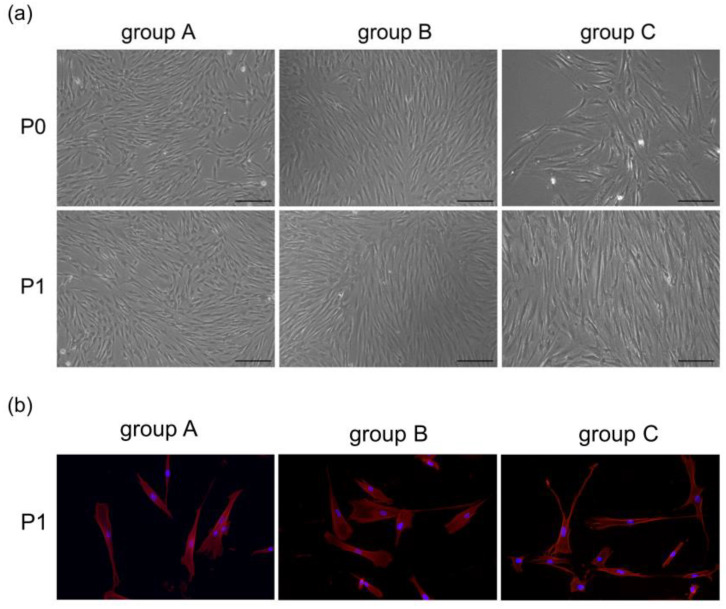
(**a**) Morphological features of hDPSCs at primary and passage 1 under an inverted microscope (scale bar: 200 μm). (**b**) Cells at passage 1 were stained with fluorophore-conjugated phalloidin (red) and DAPI nuclear counterstain (blue). The cells were spindle shape and radially arranged from the center of the tissue block. In group C, certain cells became more elongated and irregular.

**Figure 3 bioengineering-10-00012-f003:**
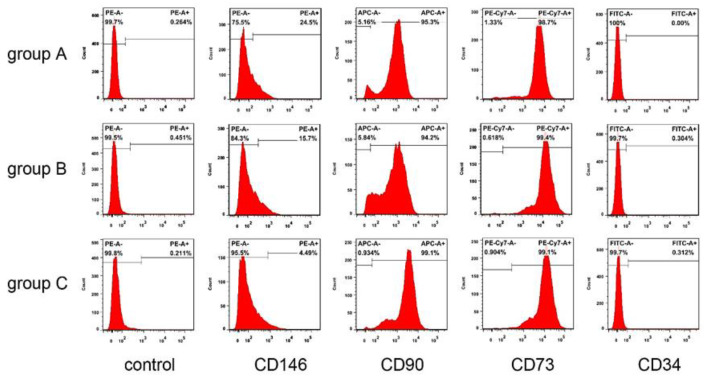
Expression of surface markers in the three groups. Flow cytometric analysis showed that hDPSCs were positive for CD73 (>90%), CD90 (>90%), with low expression of CD146 (<30%), and negative for CD45 (<1%).

**Figure 4 bioengineering-10-00012-f004:**
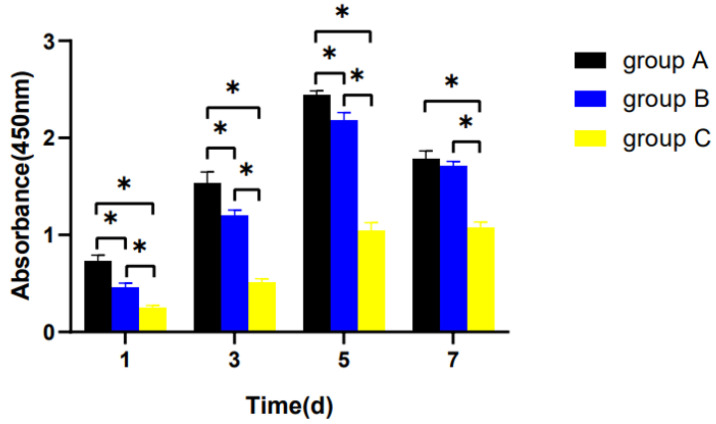
Cell proliferation in the three groups. hDPSCs were cultured with a complete culture medium for 1 week. The proliferation level was assessed with the CCK-8 assay on days 1, 3, 5, and 7. ** p* < 0.05.

**Figure 5 bioengineering-10-00012-f005:**
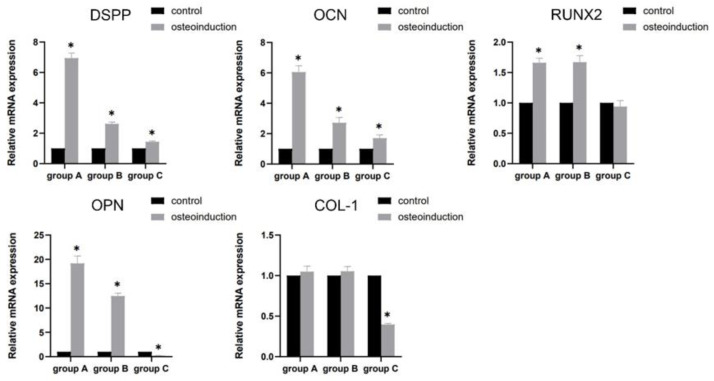
The mRNA expression of DSPP, OCN, RUNX2, OPN, and COL-1 after osteogenic induction for 7 days. ** p* < 0.05, versus control.

**Figure 6 bioengineering-10-00012-f006:**
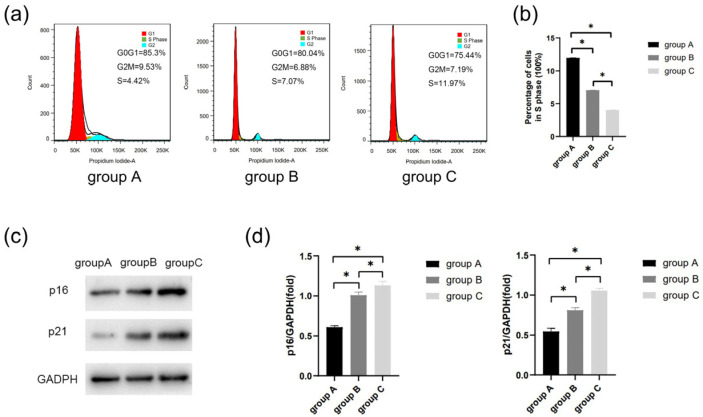
The cell senescence was determined by flow cytometric analysis and Western blot. (**a**) Cell cycle of each group was detected by flow cytometry. (**b**) Percentage of cells in S phase. (**c**) Protein expression of p21 and p16 in the three groups. (**d**) Quantitative analysis of p21 and p16 immunoblot. ** p* < 0.05.

**Figure 7 bioengineering-10-00012-f007:**
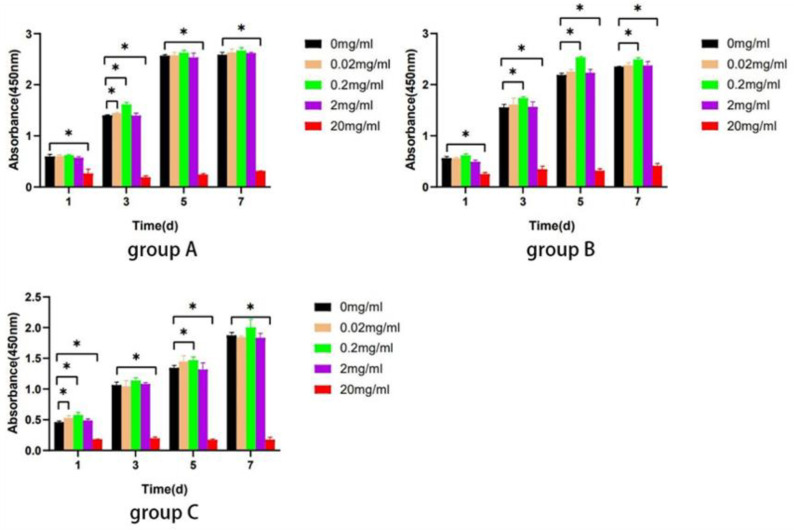
Biodentine on the proliferation ability of hDPSCs. The CCK-8 assay was performed after stimulation with different concentrations of Biodentine on days 1, 3, 5, and 7. ** p* < 0.05, versus control.

**Figure 8 bioengineering-10-00012-f008:**
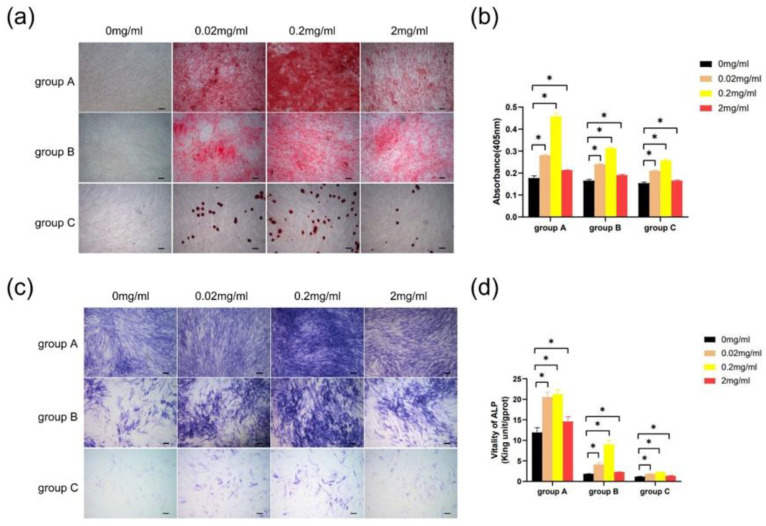
Effects of Biodentine on the mineralization of hDPSCs. (**a**) Images of alizarin red S staining (scale bar: 200 μm). (**b**) Quantitative measurement of alizarin red staining of hDPSCs cultured with different concentrations of Biodentine extracts. (**c**) Images of ALP staining (scale bar: 200 μm). (**d**) ALP activity test. * *p* < 0.05, versus control.

**Figure 9 bioengineering-10-00012-f009:**
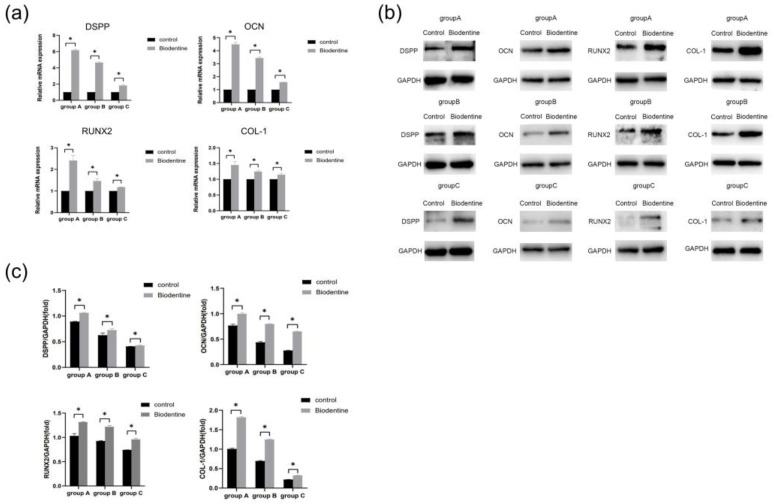
Effects of Biodentine on the odontogenic/osteogenic differentiation of hDPSCs. (**a**) The mRNA expression levels of DSPP, OCN, RUNX2, and COL-1 detected by qRT-PCR analysis on day 7. (**b**) The protein expression of DSPP, OCN, RUNX2, and COL-1 detected by Western blot analysis on day 14. (**c**) Quantitative analysis of DSPP, OCN, RUNX2, and COL-1 immunoblot. * *p* < 0.05, versus control.

**Table 1 bioengineering-10-00012-t001:** Real-time PCR primer sequences of the odontogenesis/osteogenesis-related genes.

Gene	Primer Sequence (5’–3’)
GAPDH	F:5’-GAAGGTGAAGGTCGGAGTC-3’
	R:5’-GAGATGGTGATGGGATTTC-3’
DSPP	F:5’-TGTCGCTGTTGTCCAAGAAG-3’
	R:5’-CATCACCAGAACCCTCGTCT-3’
OCN	F: 5’-CACTCCTCGCCCTATTGGC-3’
	R:5’-CCCTCCTGCTTGGACACAAAG-3’
RUNX2	F:5’-TGGTTACTGTCATGGCGGGTA-3’
	R:5’-TCTCAGATCGTTGAACCTTGCTA-3’
OPN	F:5’-CTCCATTGACTCGAACGACTC-3’
	R:5’-CAGGTCTGCGAAACTTCTTAGAT-3’
COL-1	F:5’-AAAGATGGACTCAACGGTCTC-3’
	R:5’-CATCGTGAGCCTTCTCTTGAG-3’

F: Forward, R: Reverse.

**Table 2 bioengineering-10-00012-t002:** Comparison of time required for cell attachment among groups.

Group	Time to Cell Attachment (Days)
group A	3.33 ± 0.52 ^a^
group B	5.00 ± 0.63 ^b^
group C	6.60 ± 1.14 ^c^

The time required for cell attachment was prolonged progressively with aging. There was a significant difference between the three groups. a, b, c represent significant difference when compared with other groups. *p* < 0.05.

## Data Availability

The data presented in this study are available on request from the corresponding author.

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
