# Peer review of "Effect of Biodentine on Odonto/Osteogenic Differentiation of Human Dental Pulp Stem Cells"

_bioengineering, 2022, doi:10.3390/bioengineering10010012_

Round 1

Reviewer 1 Report

This is a very interesting study.

Congratulations to the authors.

Could the authors provide a better quallity for the image c that is part of figure 5.

And also in the discussion chapter could the authors include more recent published articles.

Reviewer 2 Report

The manuscript authored by Wang et al. aimed to evaluate the effects of Biodentine on odonto/osteogenic differentiation capabilities of hDPSCs isolated from elderly. The study might provide interesting data, although deeper investigations are needed.

1.     How did the authors evaluate the cell adhesion? Particularly, a time-dependent curve would help in improving this information.

2.      Authors evaluated cell morphology only under microscopic observation. A phalloidin staining would confirm the cell morphology and possible cytoskeleton modifications.

3.     In FACS analyses authors report the expression of different MSC markers. How would the authors explain the high percentage of expression of CD146 in group A? Moreover, a cell characterization should be performed also after incubation with Biodentine.

4.     Figure 4. Immunofluorescence analyses should be performed in order to better demonstrate the osteogenic commitment of hDPSCs. To this regard, please refer to the following study: doi:10.1155/2017/3579283.

5.     In Figure 7 the authors report that 0.2 mg Biodentine is the most efficient concentration in promoting mineralized matrix deposition. How do the authors explain that group C shows, for each biodentine concentration, the lowest ALP activity and Alizarin Red staining when compared to group A and B? To this regard, it is interesting to evaluate the immunomodulatory checkpoints including Fas/FasL and PDL1. Please refer to: doi: 10.1186/s13287-021-02664-4 and doi: 10.3389/fcell.2020.00279

Minor comments:

In introduction section, a more extensive paragraph should be dedicated to the description of human dental pulp stem cell properties. Particularly, their immunomodulatory properties and differentiation abilities should be properly mentioned. See for reference doi: 10.1111/cpr.12675.

Reviewer 3 Report

Dear Authors,

I appreciated your work and the quality of presentation.

I suggest to focus the revision on some minor remarks:

Why you chose parametric analysis? The number of observations is low and statistical analysis is therefore weak.

Figure 8: please divide figures a) and c) from b). You should gain quality and intelligibility on each image. 

Best regards

Reviewer 4 Report

This is a very interesting study concerning the effect on biodentine on the increasing proliferation ability on the dental pulp cells obtained from the elder patients. The effects of biodentine on dental pulp cells were world-widely published, but there were few studies regarding the effect of biodentine on older dental pulp cells.

The authors did a lot of work mainly focusing on the aging playing an essentially determined factors contributing to dental pulp cells proliferation, differentiation, and cell senescence, but there was lack of data of biodentine treated with older pulp cells to reach the above-said function compared with younger dental pulp cells without biodentine treatment.

For example, in Figure 1, there should be a group of treatment of Group C+ biodentine over right side column.

In Figure 3, there should be a group of treatment of Group C+ biodentine

In Figure 4, there also should be a group of treatment of Group C+ biodentine.

In Figure 5 C, in western blot analysis, there should be a group of treatment of Group C +biodentine.

In Figure 7 a and c, there should be a group of treatment of Group C+ biodentine.

Round 2

Reviewer 4 Report

Now that the authors did not mainly focus on the Biodentine on odonto/osteogenic differentiation on human dental pulps from the old population, the authors should reduce the intonation of Biodentine on the pulp stem cells from the elderly shown in the title of the manuscript.
